# Super-Sensitive LC-MS Analyses of Exposure Biomarkers for Multiple Mycotoxins in a Rural Pakistan Population

**DOI:** 10.3390/toxins14030193

**Published:** 2022-03-04

**Authors:** Lei Xia, Hifza Rasheed, Michael N. Routledge, Hang Wu, Yun Yun Gong

**Affiliations:** 1School of Food Science and Nutrition, University of Leeds, Leeds LS2 9JT, UK; medlx@leeds.ac.uk; 2Pakistan Council of Research in Water Resources, Islamabad 44000, Pakistan; hifza.rasheed@pcrwr.gov.pk; 3School of Medicine, University of Leeds, Leeds LS2 9JT, UK; m.n.routledge@leeds.ac.uk (M.N.R.); umhw@leeds.ac.uk (H.W.); 4School of Food and Biological Engineering, Jiangsu University, Zhenjiang 212013, China; 5International Joint Research Laboratory of Intelligent Agriculture and Agri-Products Processing, Jiangsu Education Department, Jiangsu University, Zhenjiang 212013, China

**Keywords:** mycotoxins, biomarkers, exposure assessment, human biomonitoring, UPLC-MS/MS

## Abstract

High levels of mycotoxin contamination have been reported in various food commodities in Pakistan, however, there has been no exposure assessment study using multiple mycotoxins’ biomarkers. This study aimed to simultaneously assess the exposure to the five major mycotoxins: aflatoxin B_1_ (AFB_1_), deoxynivalenol (DON), fumonisin B_1_ (FB_1_), ochratoxin A (OTA) and zearalenone (ZEN) in a Pakistani population using an integrated approach of human biomonitoring. Human urine samples (*n* = 292) were analyzed by a super-sensitive liquid-chromatography tandem mass spectrometry (LC-MS/MS) method. Rice and wheat were also collected and analyzed for mycotoxins by the LC-MS/MS method. Food consumption data were collected using a 24 h recall method. A high prevalence of urinary AFM_1_ (66%, mean ± SD 20.8 ± 41.3 pg/mL) and OTA (99%, 134.7 ± 312.0 pg/mL) were found, whilst urinary DON, FB_1_ and ZEN levels were low. The probable daily intake (PDI) derived from the urinary biomarkers revealed that 89% of the participants had exposure to OTA exceeding the established tolerable daily intake (TDI = 17 ng/kg bw/day). The average PDI of AFB_1_ for the studied population was 43 ng/kg bw/day, with rice as the main source of AFB_1_ exposure. In summary, exposure to AFB_1_ and OTA are of health concern and require further management.

## 1. Introduction

Mycotoxins, toxic secondary metabolites produced by various fungi, contaminate many staple crops, presenting a largely ignored public health risk, especially in developing countries [1]. Mycotoxin exposure mainly occurs via ingestion of contaminated foods. Mycotoxins can be carcinogenic, nephrotoxic, immunotoxic and teratogenic for humans and animals. Moreover, mycotoxins cannot be easily removed by normal treatment such as washing the food or cooking [2]. 

Aflatoxin B_1_ (AFB_1_), deoxynivalenol (DON), fumonisin B_1_ (FB_1_), ochratoxin A (OTA) and zearalenone (ZEN) have been recognized as the major mycotoxins posing public health risks due to their toxicity and frequency of occurrence [3,4,5,6]. Simultaneous analysis of multiple mycotoxins has been developed in the last decade providing increasing evidence of the co-occurrence of these mycotoxins. These five mycotoxins and their derivatives were reported to be the predominant mycotoxins in previous co-occurrence studies [7,8]. To facilitate accurate exposure risk analysis, the research focus has moved from broad scope methods to ‘dedicated’ methods covering a narrower range of key multiple contaminants in order to improve the sensitivity [9]. This research aims to optimize the LC-MS/MS method to achieve maximum sensitivity for the five major mycotoxins including their main metabolites, and to apply this to human exposure analysis. 

Due to the limited research capability in developing countries like Pakistan, lack of co-exposure data and lack of regulatory legislation raise major food safety questions in these populations, requiring urgent research and public health attention. Therefore, focused research on major mycotoxins is a good starting point to assess the exposure of mycotoxins in countries where exposure data are limited.

Compared to exposure assessments using a food intake approach, the human biomonitoring (HBM) approach considers all routes of exposure, variations in individual’s metabolisms and lifestyle. Consequently, the assessed exposure is expected to be more closely related to the health impact of the mycotoxins [2]. Urinary aflatoxin M_1_ (AFM_1_) and urinary total DON are commonly used sensitive biomarkers to assess the exposure of AFB_1_ and DON, respectively [10]. For FB_1_, urinary free FB_1_ has been shown to correlate well with dietary intake [11]. The pharmacokinetics for OTA and ZEN are not well characterized, so urinary biomarkers for these mycotoxins need further validation. For ZEN, the existing evidence suggests that urinary excretion of ZEN, α-ZEL and possibly β-ZEL is sufficiently high for them to be used as biomarkers of exposure for ZEN and they have been used extensively in recent ZEN exposure assessment studies [12]. Overall, a good correlation was reported between urinary OTA levels and dietary OTA intake [13], despite the low excretion rate of OTA in urine.

Mycotoxin contamination has been reported in many food commodities in Pakistan, including rice, wheat, maize, milk, nuts, spices, tea etc. with frequent detection of AFB_1_ and OTA [14,15,16,17,18,19,20,21]. However, the food contamination data for other mycotoxins are scarce and there was no exposure assessment for multiple mycotoxin exposure conducted in Pakistan using the HBM approach. To bridge this gap, this paper aims to assess the exposure of AFB_1_, DON, FB_1_, DON and ZEN using the HBM approach.

## 2. Results

### 2.1. Method Validation

The limit of detection (LOD) and limit of quantification (LOQ) of the food mycotoxin analysis method and 18 batches of wheat and rice quality controls (QCs) are summarized in Table 1. The method shows inter-day and intra-day reproducibility for all the screened species with acceptable CV%. The intra-day variation was measured by three consecutive injections of the standards on the same day, whereas the inter-day variation was measured by injections of standards (same concentrations as the ones used for setting-up the standard curves) on three consecutive days. As shown in the table, all of the analyzed mycotoxins showed good stability and apparent recoveries (R_A_, around the range of 80–120%) at spiked concentration levels according to the guidelines [22]. The matrix effect for the analytes was low, with values of spiked samples in the extractant matrix ranging from 81–103% of the spiked concentration (see Appendix A). Only standard curves with linearity ≥0.995 were considered for the analysis. The coefficient of variance (CV%) of the 26 QCs illustrated a satisfactory stability of the method. In all, 10% (*n* = 26) of the total sample size was repeated from the extraction stage. No significant difference (*p* > 0.05) was observed for the screened analyte levels between the original extracted sample and repeats for both the rice and wheat mycotoxin analyses (*n* = 26).

The parameters used to validate the urinary mycotoxin biomarker analysis method are summarized in Table 2. The method shows inter-day and intra-day reproducibility for all the screened species with an acceptable CV%. The inter-day and intra-day variation was measured in the same way as the food samples. The QC samples also showed reproducible (CV% ≈ 10%) results with acceptable R_A_. The matrix effect for the analytes was low, with the matrix values ranging from 77–99% of the spiked concentration, (see Appendix A). Only standard curves with linearity ≥0.995 were considered for the analysis. For the repeated samples, no significant difference (*p* > 0.05) was observed for the screened analyte levels between the original extracted sample and repeats for urinary mycotoxin biomarker analysis (*n* = 31), apart from urinary DON levels. The significant difference observed for urinary DON biomarker level can be accounted for by the low level and low occurrence of this biomarker. 

The urinary AFM_1_ and total DON concentrations were also determined in our previous study for most of the urine samples (*n* = 264) using a single mycotoxin analytical method [23]. A paired sample *t*-test was performed, and no significant difference (*p* > 0.05) was observed between the values from the two studies.

### 2.2. Demographic Characteristics of the Participants

Demographic data are summarized in Table 3. In all, 60% of the studied participants were males. The average age of the participants was 36.6 ± 17.4 years (range 4–80 years). Village Chak-46 from Sahiwal had the highest number of participants (35%). The majority (90%) of the participants were from families participating in farming activities, including farmers, housewives and their children. Wheat chapatti was the staple food, as 99% of the participants consumed chapattis on the previous day, with males’ consumption higher than that of females (389.1 g/day vs. 321.5 g/day, *p* < 0.001). In contrast, rice was consumed less frequently (42.5%) in this population and no significant difference in rice consumption was observed between genders.

### 2.3. Mycotoxin Contamination in Rice and Wheat

Table 4 summarizes the concentrations of the major mycotoxins found in rice and wheat samples that exceeded 10% of the total sample size. For rice, the most frequently detected mycotoxin was ZEN, which contaminated 100% of the rice samples with average concentrations of 5.2 ± 3.2 µg/kg. No significant difference was observed for ZEN concentration in rice samples from different villages (*p* = 0.170). The second most frequently detected mycotoxin was AFB_1_, which was found in 66% of the rice samples with an average concentration of 7.6 ± 14.06 µg/kg. A significant difference (*p* = 0.017) was observed for the AFB_1_ contamination level in rice across the villages. FB_1_ and OTA were only found in three (5%), and one (2%) of the rice samples and none of the rice samples had a detectable concentration of DON. For wheat, the most frequently detected mycotoxin was also ZEN. However, compared to rice, the positive rate (33%) and contamination concentration (1.31 ± 0.92 µg/kg) of ZEN in wheat is much lower. The difference of ZEN in wheat in different villages was found to be significant (*p* = 0.047). AFB_1_, FB_1_ and OTA were only found in three (2%), one (0.5%) and ten (5%) of the wheat samples and none of the wheat samples had a detectable concentration of DON.

### 2.4. Urinary Mycotoxin Biomarkers

All the urine samples were analyzed by the super-sensitive multiple urinary mycotoxin biomarker LC-MS/MS method. Almost all the 292 urine samples (99%) had detectable concentrations of urinary mycotoxin biomarkers. The details of mycotoxin biomarker concentration by village and in the total population are summarized in Table 5. Samples with levels below the LOD were assigned with concentrations of ½ LOD for statistical analysis. Overall, OTA was the most frequently detected mycotoxin, at 99%, and a concentration mean of 134.7 ± 312.0 pg/mL. FB_1_ was detected in 77% of samples with an average concentration of 17.4 ± 26.8 pg/mL. AFM_1_, the urinary biomarker for AFB_1_, was detected in 66% of the samples and the average concentration was found to be 20.8 ± 41.3 pg/mL. DON and DOM-1 were found in 35% and 21% of the total urine samples with averages of 138.4 ± 266.5 and 51.8 ± 107.0 pg/mL, respectively. In contrast to the high contamination prevalence in food samples, ZEN biomarkers had the lowest occurrence among the studied mycotoxins. ZEN, α-ZEL and β-ZEL were detected in 37%, 27% and 38% with average concentrations of 16.7 ± 44.1, 10.5 ± 14.4 and 20.0 ± 32.0 pg/mL, respectively.

Overall, significant differences (*p* < 0.001, apart from α-ZEL with a *p* value of 0.002) were observed for all of the mycotoxins across the five villages. Participants from village Chak-46 and BP showed higher urinary AFM_1_ and OTA concentrations compared to the other villages. Participants from village Chak-46 also showed the highest concentrations of both AFM_1_ and OTA in their urine, with average concentrations of 36.9 ± 58.1 and 219.3 ± 505.4 pg/mL, respectively. For urinary DON biomarkers, participants from villages 50 and BB showed the highest concentrations of DON in their urine samples, although the levels were overall very low. No significant correlation (*p* = 0.12) was observed between the urinary DON level and DOM-1 level. For urinary FB_1_, urine samples from villages BB and BP showed higher average concentrations than the other villages. For urinary ZEN biomarkers, village 50 showed the highest concentrations of ZEN and α-ZEL in their urine samples, whereas participants from village BB showed the highest concentrations of β-ZEL in their urine samples. A significant correlation (*p* = 0.08) was observed between the urinary ZEN and α-ZEL concentrations, but no significant (*p* = 0.23) correlation was observed between urinary ZEN and β-ZEL concentrations.

The urinary AFM_1_ level was higher in males than in females (25.6 ± 48.7 vs 13.6 ± 25.1 pg/mL, *p* = 0.02); the same trend was also found for urinary FB_1_ (18.4 ± 25.6 vs 15.8 ± 28.5 pg/mL, *p* = 0.02), and urinary ZEN (18.9 ± 53.6 vs 13.5 ± 23.5 pg/mL, *p* < 0.001). However, the significant difference in urinary AFM_1_ and ZEN concentrations between genders vanished after considering ‘village’ as a confounder. No significant correlation was observed between the urinary mycotoxin biomarker concentrations with age, nor with rice or wheat food consumption.

### 2.5. Co-Exposure of Mycotoxins Measured by Urinary Biomarkers

Co-exposure of the mycotoxins was common (Figure 1) in the studied population, and 82% of the urine samples tested positive for at least three mycotoxin biomarkers. There were 11 urine samples (3.8%) that tested positive for seven different mycotoxin metabolites. The most common co-exposure mycotoxins were FB_1_ and OTA, which were found in 220 (75%) urine samples, followed by the combination of OTA and AFM_1_ found in 190 (65%) samples. For multiple mycotoxin exposure, the combination of AFM_1_, OTA and FB_1_ was the most common and was found in 142 (49%) samples.

### 2.6. Estimated Dietary Mycotoxin Exposure

The PDI estimated from the biomarker concentrations (deterministic) for all the studied mycotoxins are summarized in Table 6. For AFB_1_, the average and maximum PDI estimated by the urinary AFM_1_ concentration was 0.043 and 0.852 µg/kg bw/day, respectively. In total, 259 (89%) of the studied population had PDI for OTA higher than the established TDI (0.017 µg/kg bw/day) [25]. The maximum PDI for DON (0.153 µg/kg bw/day), FB_1_ (1.741 µg/kg bw/day) and ZEN (0.214 µg/kg bw/day) were all lower than the established TDI (DON: 1 µg/kg b.w./day; FB_1_: 2 µg/kg b.w./day; ZEN: 0.25 µg/kg b.w./day), as defined by the Joint FAO/WHO Expert Committee on Food Additives [26,27] or European Food Safety Authority [28].

## 3. Discussion

### 3.1. Biomarker Levels and Possible Sources of Mycotoxin Exposure

In this study, the levels of urinary biomarkers of five major mycotoxins (AFB_1_, DON, FB_1_, OTA and ZEN) have been assessed in people from the Punjab province in Pakistan. The biomarker for OTA suggested a high frequency of positive samples and high exposure levels in this population. However, the foods (rice and wheat) had very low levels of OTA detected. This disparity suggested the exposure is from a food source not analyzed in this study. A previous study also reported a low contamination rate (6%) of OTA in rice samples collected in the same region [14]. Similar levels of urinary OTA were reported in Bangladesh [34]. Both Pakistani and Bangladeshi populations consume curries containing chili and various spices on a daily basis. A high concentration of OTA has been found in Pakistan chili and other spices [35], which may be one of the major contributors to OTA exposure. 

AFM_1_ was found in 66% of the urine samples, with an average concentration of 21 ± 41 pg/mL and maximum concentration of 421 pg/mL. The detected urinary AFM_1_ concentration is comparable to that reported in high aflatoxin exposure for populations in Nigeria (73%, average 40 pg/mL, maximum 620 pg/mL) and Tanzania (86%, average 37 pg/mL, maximum 2840 pg/mL) [7,36]. In Asia, the Bangladeshi population had a lower positive rate (8%) and maximum concentration of 120 pg/mL [37]. The high AFB_1_ levels of the rice samples reported in our study suggests that rice consumption is an important contributor to AFB_1_ exposure. It is possible that other food commodities such as cereals, spices, black tea and milk consumed frequently by the studied population may also contribute to the exposure as a number of studies from this region showed such foods were heavily contaminated by AFB_1_ [15,16].

We found very low urinary DON concentration (mean 138.4 ng/mL, median nd) in this population compared to most previously reported studies [38]. This low level of urinary DON is consistent with the low level of DON contamination in the foods of this population. This is likely because the fungi producing DON favors a mild temperate climate. No significant correlation was found between urinary DON and DOM-1 concentrations. This can be partially attributed to the low levels of DON and DOM-1 in the studied population, and supports previous suggestions that DOM-1 may not be a suitable biomarker to assess human DON exposure [39].

Urinary FB_1_ was detected in 77% of the urine samples, however the levels were low. This showed our analytical method is super-sensitive for FB_1_ detection. Urinary FB_1_ concentration in this study is lower compared to most published biomonitoring data such as from Cameroon (average 2960 maximum 4800 pg/mL), Nigeria (average 1090 maximum 1488 pg/mL) and European populations (median 3230 maximum 5430 pg/mL) [7,40,41]. The low concentration of the urinary biomarker is consistent with the low FB_1_ contamination of rice and wheat reported here and elsewhere [14]. Maize is the major crop that is responsible for FB_1_ exposure [42], and this population has a relatively low maize consumption. In addition, the hot temperature during summertime in Pakistan is not suitable for the growth of FB_1_ producing fungi.

Similar to DON, the urinary ZEN level in this population was lower than most published biomonitoring studies for ZEN [38], indicating that the studied population has a low risk of ZEN exposure. This agrees with the low concentrations of ZEN found in rice and wheat samples. The average levels of ZEN, α-ZEL and β-ZEL were not much different from each other. No significant correlation was found between urinary ZEN, α-ZEL and β-ZEL. The correlation between ZEN and its metabolites in urine is not consistent in different biomonitoring studies [7,38].

### 3.2. Exposure Risk Assessment for Mycotoxins

A high prevalence of AFB_1_ (66%) with a high urinary AFM_1_ concentration was observed for the studied population. As AFB_1_ has been classified as Group 1 carcinogen, there is no threshold for safe exposure level, so this population is at high exposure risk. The average PDI for AFB_1_ (43 ng/kg bw/day) is comparable to those countries with frequent AFB_1_ contamination detection, such as: Brazil (60 ng/kg bw/day) and Zimbabwe (48/56 ng/kg bw/day) [43,44], but lower than those countries that have had aflatoxicosis outbreaks, such as Kenya (353 ng/kg bw/day).

OTA was detected in almost all the urine samples. Although OTA is also a possible carcinogen, the evidence of carcinogenicity for OTA exposure in humans is inadequate, therefore it is a Group 2B carcinogen with a TDI of 17 ng/kg bw/day. The estimated PDI for OTA of the cohort exceeded the established TDI for 259 (89%) participants, which suggests a high exposure of OTA in the studied population. The PDI of FB_1_ is well below the TDI set by JECFA (JECFA, 2007). The PDI levels of DON and ZEN are both much lower than the established TDI, which agreed with the low contamination found in the wheat and rice samples. 

Mycotoxin co-contamination in the diet is concerning because existing knowledge of the human health risks of mycotoxins’ co-exposure is limited. Here, 82% of the population were exposed to at least three mycotoxins and the combination of the three most frequently detected biomarkers AFM_1_ + FB_1_ + OTA were found in 49% of the population. To date, health-based guideline values are not available regarding the co-exposure of mycotoxins. However, several in vitro studies have illustrated either additivity or synergic cytotoxicity, and/or genotoxicity for the exposure of AFM_1_ + FB_1_ + OTA, in liver and kidney cell lines [45,46]. 

### 3.3. Influence of Geographic and Demographic Factors on Mycotoxin Exposure

Overall, AFB_1_ and OTA exposure were common, whilst DON, FB_1_ and ZEN showed relatively low exposure in this population. This observation can be accounted for by the influence of climate conditions on different toxigenic fungi species, as well as by dietary habits in Punjab, Pakistan. Both AFB_1_ and OTA are mainly produced by *Aspergillus* fungi that prefer hot humid conditions, whereas DON, FB_1_ and ZEN are mainly produced by *Fusarium* fungi that thrive in temperate conditions. The average temperature in the studied locations of 37 °C [47], is suitable for the growth of *Aspergillus* species but can be too high for the growth of *Fusarium* species [48,49]. In addition, significantly higher exposure of AFB_1_ and OTA were observed for participants from village 46 and BP, both located in the north of the Punjab province. This observation, also reported by [14], can be explained by the difference in humidity between the south and north of the Punjab Province. According to Khattak and Ali [50], the northern Punjab receives more rainfall compared to the south and higher humidity also promotes the growth of *Aspergillus* fungi during grain storage. There were no significant differences in PDI for AFB_1_ and OTA according to gender which is in agreement with other reports: AFM_1_ [37]; OTA [51].

### 3.4. Limitation and Uncertainties

PDIs derived from the urinary biomarker concentrations were used for risk assessment. A number of factors contributed to the uncertainty of the data. First, the urine excretion volume used in deriving the PDI was obtained by using the default assumption, which can be improved by measuring the 24 h urine excretion volume for each of the participants. In addition, the urine excretion rate used to derive the PDI for AFB_1_, FB_1_, OTA and ZEN has not yet been fully validated for humans, requiring more toxico-kinetic data to confirm the exact urine excretion rate for those mycotoxins, particular for ZEN, for which the data were only obtained from one adult male [33]. Furthermore, the low excretion rate of AFB_1_, FB_1_, OTA and ZEN in the urine can greatly increase the uncertainty in deriving the corresponding PDIs. The limitations of the study include the lack of accurate estimates of food-based mycotoxins’ daily intake due to the crude food consumption data and the small variety of food sampled for mycotoxin analysis. This makes the comparison of food-based and biomarker-based exposure estimates difficult.

## 4. Conclusions

Using super-sensitive LC-MS/MS methods for analysis of multiple mycotoxins in urine, we found that populations from the Punjab region of Pakistan had a high prevalence of AFB_1_ and OTA exposure. The estimated AFB_1_ and OTA exposure levels were comparable to those countries reported with frequent AFB_1_ and OTA contamination in food, such as Nigeria and Tanzania. Although rice was the major source of exposure for AFB_1_, neither rice nor wheat appeared to be the main source of exposure to OTA. Other potential sources of mycotoxin contamination in food should be investigated. Urinary biomarker analysis showed that exposure risks to DON and ZEN in this population are low. The high exposure identified for carcinogenic AFB_1_ and OTA is of public health concern for Pakistan, especially as there is a lack of regulatory enforcement for aflatoxin and OTA there. Co-exposure to mycotoxins AFB_1_ + OTA + FB_1_ was frequently observed for the studied population. Regional differences in AFB_1_ and OTA exposure were observed between northern and southern Punjab provinces. Our results should be of value to guide effective intervention strategies to reduce mycotoxins’ exposure for public health protection in Pakistan. 

The approach used in this study is effective in conducting exposure assessment and risk management in areas where no/limited data for mycotoxin exposure are available. The highly sensitive analytical method allows accurate exposure measurement for key mycotoxins providing valid data for health risk studies. Moreover, by analyzing the crops that are highly consumed in the selected population and susceptible for mycotoxin contamination, it helped to quickly identify the source of exposure for the mycotoxins of concern, and the risk manager will then be able to make effective interventions to control the exposure risk.

## 5. Materials and Methods

### 5.1. Reagents and Chemicals

Acetonitrile (ACN; LC-MS grade), acetic acid (HPLC grade) and formic acid (LC-MS grade) were purchased from Fisher Scientific (Loughborough, UK). Methanol (LC-MS grade) was purchased from SLS (Nottingham, UK). Mycotoxin standards AFB_1_ and AFM_1_ (≥99.5%) were purchased from Romer Labs (Runcorn, UK). Other mycotoxins’ standards DON, DOM-1, FB_1_, OTA, ZEN, α-ZEL and β-ZEL (≥99%) were purchased from Sigma-Aldrich (Poole, UK). Internal standard (IS) ^13^C_17_-AFM_1_ (≥99%) was purchased from Romer Labs (Runcorn, UK). Internal standards ^13^C_15_-DON, ^13^C_34_-FB_1_ and ^13^C_20_-OTA (≥99%) were purchased from Sigma-Aldrich (Poole, UK). β-glucuronidase from *E. coli* was purchased from Sigma-Aldrich (Poole, UK). All the standards and IS were dissolved in ACN, apart from FB_1_ and ^13^C_34_-FB_1,_ which were dissolved in 1:1 ACN: H_2_O and stored at −20 °C. The Oasis HLB Prime (3 cc, 60 mg) cartridge was purchased from Waters (Wilmslow, UK). The 0.22 µm PTFE syringe filter was purchased from Phenomenex (Macclesfield, UK). 

A multiple mycotoxin standard containing 200 ng/mL DON, 20 ng/mL AFB_1_, FB_1_, OTA, ZEN were prepared from the standards using ACN and used for rice/wheat analysis. The highest standard solution was prepared by diluting working standard solution with matrix-matched solution at a dilution factor of 1:1 (*v*/*v*). The matrix-matched dilution solution was obtained by extracting 2 g of the blank rice/wheat samples with 8 mL of the extraction solution (79% acetonitrile ACN, 20% water with 1% acetic acid) followed by a (1:7) dilution by LC-MS grade water. Other standard solutions were prepared by diluting the highest standard solution 2, 5, 10, 20, 50, 100, 200, 500, 1000 times. 

A multiple mycotoxin working standard solution, containing 2 µg/mL DON, 200 ng/mL AFM_1_, DOM-1, FB_1_, OTA, ZEN, α-ZEL and β-ZEL, was prepared from the standards using ACN and used for urine analysis. A multiple mycotoxin working IS solution containing 5 ng/mL ^13^C_17_-AFM_1_, 10 ng/mL ^13^C_34_-FB_1_ and ^13^C_20_-OTA, 25 ng/mL ^13^C_15_-DON was prepared from the IS solutions, using ACN for urine analysis. The highest standard solution was prepared by diluting working standard solution with dilution solvent (1:9 ACN: H_2_O, *v*/*v*) at a dilution factor of 1:39 (*v*/*v*). Other standard solutions were prepared by diluting the highest standard solution 2.5, 5, 10, 25, 50, 100, 250, 500, 1000, 2500 times. 20 µL of the IS working solution were added to the standard solutions before injection. 

### 5.2. Study Population and Sample Collection

The study was extended from a previous study assessing arsenic exposure, in which 395 participants were recruited from six villages in four districts as shown in Figure 2: Chak-46, 48 and 49 from district Sahiwal (SW); Badarpur (BP) from district Kasur, Basti Balochan (BB) from Bahawalpur; and Kotla Arab (KA) from Rahim Yar Khan district in the Punjab province, Pakistan [52]. Villages Chak-48 and 49 were grouped with Chak-50 due to the geographical proximity and small sample size.

The demographic and health status data for each participant were obtained using a questionnaire. The consumption data of rice and chapatti (wheat based staple food) were obtained using a 24 h dietary recall method [52]. 

Wheat and rice were sampled from the households during May to July 2014. Raw packed rice samples (*n* = 105) were collected from local shops (due to no local production in the selected area), whereas wheat consumed in the villages was cultivated locally. Wheat grain samples (*n* = 195) from two of the most cultivated wheat varieties were collected from the households of six villages. Individual samples (150 g each) were pooled into eight composite samples weighing in the range of 0.9–7.5 kg. For raw rice and wheat samples, sterile re-sealable airtight polyethylene zip lock bags were used. After collection, raw rice (250 g) and wheat samples (150 g) were stored at room temperature, then were shipped to the University of Leeds, UK, by FedEx courier with dry ice under strict quarantine regulations and stored at −20 °C prior to analyses. As the raw rice and wheat samples have been used for another study, only samples with weight of ≥10 g were analyzed in this study which led to 62 raw rice samples and 195 raw wheat samples. 

Spot urine samples were also collected during May to July 2014 for biomarker analysis. Among the 395 urine samples, only 292 of them had sufficient volume (≥5 mL) left for the urine analysis. Available urine samples (*n* = 292, age 4–80 years) were shipped to Leeds on dry ice and stored at −20 °C until analysis of urinary mycotoxin biomarkers, thus in the study we considered the 292 subjects as study participants. Informed consent was obtained from all participants. Ethical approval was granted from the University of Leeds Ethical Committee (MEEC 17-036) and the National Bioethics Committee Pakistan (4-87/14/NBC-150/RDC/3). 

### 5.3. Food Mycotoxin Analysis

The extraction method was adopted with minor modification of Juan, Covarelli [53]. In brief, 2 g of raw rice or wheat samples were extracted with 8 mL of the extraction solution (79% acetonitrile ACN, 20% water with 1% acetic acid) with 2 h of shaking under 2500 rpm at room temperature. After the extraction, the mixture was centrifuged (4 °C, 5000× *g*) for 20 min to separate the solid contents. 125 µL of the supernatant was diluted by 875 µL of LC-MS grade water and then filtered through a 0.22 µm PTFE syringe filter. Rice/wheat mycotoxin levels were measured using a Thermo Vanquish Flex binary UPLC system coupled with Thermo TSQ Quantiva triple Quadrupole mass spectrometer using ESI. The details of the UPLC-MS/MS parameters are in the Appendix A. The mycotoxin levels were quantified in the multiple reaction monitoring (MRM) mode and the method parameters are summarized in Appendix A. The rice/wheat mycotoxin concentrations were quantified using Quan Browser built-in Xcalibar 4.1 software by comparing the peak area to an external standard curve.

### 5.4. Urinary Mycotoxin Biomarker Analysis

The extraction method was adopted, with minor modifications, from Sarkanj et al. (2018). In brief, 1 mL of the centrifuged (15 min, 4 °C, 5000× *g*) urine sample was spiked with 20 µL of the IS working solution, before dilution with 1 mL of PBS. After overnight digestion at 37 °C with 5750 units of β-glucuronidase, the urinary mycotoxin biomarkers were extracted by the Oasis HLB Prime (3 cc, 60 mg) cartridge. Following pre-conditioning, the urine sample were loaded on the column and then washed with 2 mL of water. Samples were eluted in 2 mL ACN and 2 mL 0.1% formic acid in ACN (to help to elute FB_1_ and OTA) solution. Samples were dried overnight under vacuum at room temperature before being reconstituted in 1 mL of ACN:water (*v*/*v*, 1:9). Urinary mycotoxin biomarker levels were measured by a Waters Acquity UPLC I-Class system coupled with Waters TQ-XS tandem MS using ESI. The details of the UPLC-MS/MS analysis are in the Appendix A. The mycotoxin levels were quantified in the MRM mode and the method parameters are summarized in Appendix A. The urinary mycotoxin biomarker concentrations were quantified using Targetlynx XS built in Masslynx v4.2 software. The urinary DOM-1, ZEN, α-ZEL and β-ZEL levels were determined by comparing the peak area to an external standard curve, whereas urinary AFM_1_, DON, FB_1_, OTA levels were corrected by the signal of the IS.

### 5.5. Validation of Analytical Method 

For the food mycotoxin analysis, QC samples (one rice and one wheat), were included in each batch of 18 samples. The QC samples were prepared by spiking 20 µL of a stock solution in ACN to give a final concentration of 3.2 µg/kg AFB_1_, 160 µg/kg DON, 48 µg/kg ZEN, 48 µg/kg FB_1_ and 9.6 µg/kg of OTA. The spiked rice/wheat QC samples were left in 4 °C fridge overnight before analysis. These QCs were used to evaluate the method apparent recovery (R_A_), reliability and variation of the method. Apparent recovery (R_A_) was calculated using the following Equation (1):(1)$$R_{A}\;\left(\%\right)=\frac{\mathrm{Average}\;\mathrm{area}\;\mathrm{of}\;\mathrm{spiked}\;\mathrm{samples}}{\mathrm{Average}\;\mathrm{area}\;\mathrm{of}\;\mathrm{solvent}\;\mathrm{standards}}$$

10% of the total sample size (*n* = 26, 6 rice samples, 20 wheat samples) were randomly selected and extracted again to further validate the method. 

For the urinary biomarker analysis, the guidance for method validation from EU Commission Decision 2002/657/EC [7,54] was followed. Limit of detection (LOD), limit of quantification (LOQ), repeatability, reproducibility, trueness and linearity were evaluated. Recovery experiments were performed by spiking the blank urines with mycotoxin working standard solutions with the same concentration as the established calibration curve. The trueness and selectivity were estimated by recovery. The intra-day and inter-day variation were evaluated from measurements repeated on three days with six determinations per concentration level. LOD and LOQ were determined based on a signal to noise ratio of 3:1 and 10:1 from spiked urine sample by using Targetlynx S/N. For AFM_1_, DON, FB_1_ and OTA, the peak area ratio was used to determine the concentration, whereas other mycotoxins and biomarkers used peak area for all the calculation. 

For sample analysis, two QC samples, spiked with the working multiple mycotoxin standard (1:10 and 1:100) and 20 µL of multiple mycotoxins IS were include in each batch of 14 samples. Then 10% of the total sample size (*n* = 31) was extracted again to further validate the method.

### 5.6. Dietary Mycotoxin Intake Estimation

The probable daily intake (PDI) was derived using Equation (2) [8] from urinary biomarker concentrations using both deterministic and probabilistic approaches.
(2)$$\mathrm{PDI}=\frac{\mathrm{Urinary}\;\mathrm{biomarker}\;\mathrm{concentration}\;\times \mathrm{daily}\;\mathrm{urine}\;\mathrm{excration}\;\mathrm{volume}}{\mathrm{body}\;\mathrm{weight}\;\times \mathrm{urinary}\;\mathrm{biomarker}\;\mathrm{excretion}\;\mathrm{rate}}$$

Daily urine volumes were assumed to be 36 and 18 mL/kg bw/day for children (4–9 years) and adolescents (10–19 years), and 2000 mL/day for adult males and 1600 mL/day for adult females, respectively, as recommended by [55]. Individual body weight is available from the survey. The references for the urinary excretion rate of the mycotoxin urinary biomarkers are as follows: AFM_1_, 1.3% [29]; DON, 70% [30]; FB_1_, 0.3% [31]; OTA, 2.5% [32]; ZEN, 9.4% [33]. The exposure of mycotoxins was assessed by comparing the PDI level to the health-based guidance values (TDI or PMTDI) level set by either JECFA or EFSA. Above those health-based guidance values, the exposure to the mycotoxin is regarded to pose public health concern.

### 5.7. Statistical Analysis

Means (±SD), medians and range were used to summarize the levels of the mycotoxin biomarkers. For levels below the LOD a value of 1/2 LOD was assigned for statistical calculation [10]. The urinary biomarker concentrations were log-transformed for all statistical analysis as the data showed skewed distributions. Linear correlation and regression analysis were performed for the association between urinary mycotoxin biomarker level and food consumption data. Chapatti and rice consumption data were adjusted by participants’ body weight. Mann–Whitney or Kruskal–Wallis test was used to analyze the urinary mycotoxin biomarker levels in relation to various demographic data. All analyses were carried out using IBM SPSS Statistics Version 26 and a *p*-value of less than alpha 0.05 was used to assign for statistical significance. 

## Figures and Tables

**Figure 1 toxins-14-00193-f001:**
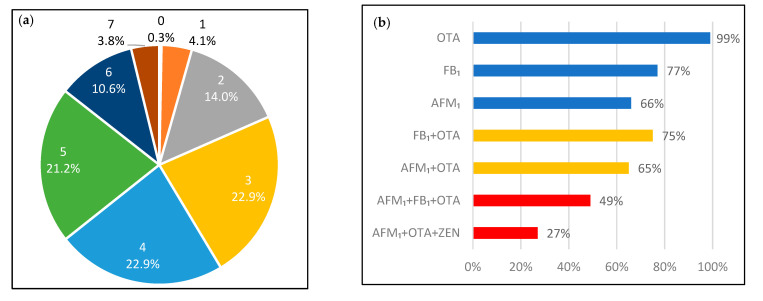
(**a**). Distribution of the participants regarding the number of co-exposed mycotoxins by percentages. The number above the percentages illustrate the number of co-occur mycotoxin biomarkers found in the urine samples; (**b**). The % for the most common combination: single mycotoxin; two mycotoxins; and three mycotoxins co-exposure.

**Figure 2 toxins-14-00193-f002:**
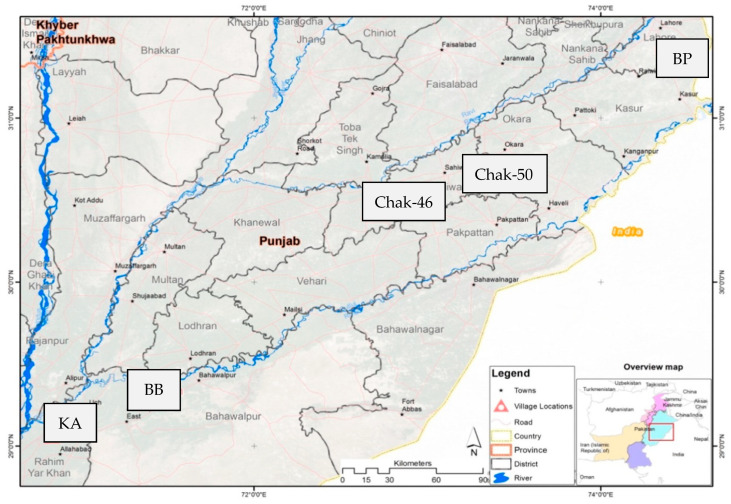
Geographic location of the studied cohort, reproduced from [52] with permission from Elsevier, 2022.

**Table 1 toxins-14-00193-t001:** Method validation parameters for food mycotoxin analysis.

Analyte	Calibration Range (µg/kg)	Intra-Day (*n* = 3) CV (%)	Inter-Day (*n* = 3) CV (%)	Matrix LOD (µg/kg)	Matrix LOQ (µg/kg)	QC (*n* = 26) CV (%)	QC (*n* = 26) R_A_ (%)
AFB_1_	0.16–320 ^a^	4	3	0.064	0.16	9	89
DON	64–3200 ^b^	4	2	32	64	8	89
FB_1_	1.6–320 ^c^	5	4	0.64	1.6	9	107
OTA	0.8–320 ^d^	5	3	0.32	0.8	14	90
ZEN	1.6–320 ^c^	4	4	0.64	1.6	9	90

LOD—Limit of detection; LOQ—Limit of quantification; QC—quality control; CV—coefficient of variance; R_A_—apparent recovery; the matrix LOD and LOQ are for both rice and wheat matrixes; Calibration points are as follows: ^a^ 0.16, 0.32, 0.64, 1.6, 3.2, 6.4, 16, 32, 64, 160, 320 µg/kg; ^b^ 64, 160, 320, 640, 1600, 3200 µg/kg; ^c^ 1.6, 3.2, 6.4, 16, 32, 64, 160, 320 µg/kg; ^d^ 0.8, 1.6, 3.2, 6.4, 16, 32, 64, 160, 320 µg/kg.

**Table 2 toxins-14-00193-t002:** Method validation parameters for urinary mycotoxin biomarkers analysis.

Analyte	Calibration Range (ng/mL)	Intra-Day (*n* = 3) CV (%)	Inter-Day (*n* = 3) CV (%)	Matrix LOD (ng/mL)	Matrix LOQ (ng/mL)	QC (*n* = 22) CV (%)	QC (*n* = 22) R_A_ (%)
AFM_1_	0.002–5 ^a^	5	4	0.001	0.002	10	104
DON	0.2–50 ^b^	4	6	0.1	0.2	10	101
DOM-1	0.05–5 ^c^	4	3	0.02	0.05	11	92
FB_1_	0.002–5 ^a^	4	4	0.001	0.002	13	111
OTA	0.002–5 ^a^	2	2	0.001	0.002	11	96
ZEN	0.02–5 ^d^	4	12	0.01	0.02	9	127
α-ZEL	0.02–5 ^d^	5	12	0.01	0.02	11	106
β-ZEL	0.02–5 ^d^	4	12	0.01	0.02	9	78

CV—coefficient of variance; LOD—Limit of detection; LOQ—Limit of quantification; QC—quality control. Calibration points are as follows: ^a^ 0.001, 0.002, 0.005, 0.01, 0.02, 0.05, 0.1, 0.2, 0.5, 1, 2, 5 ng/mL; ^b^ 0.1, 0.2, 0.5, 1, 2, 5, 10, 20, 50 ng/mL; ^c^ 0.02, 0.05, 0.1, 0.2, 0.5, 1, 2, 5 ng/mL; ^d^ 0.01, 0.02, 0.05, 0.1, 0.2, 0.5, 1, 2, 5 ng/mL.

**Table 3 toxins-14-00193-t003:** Demographic characteristics of the studied population.

Characteristics	Chak-46	Chak-50	BB	BP	KA	All Villages
Total n	101	74	35	23	59	292
Gender ***						
Male n, (%)	65 (64) ^b^	63 (85) ^a^	12 (34) ^c^	14 (61) ^b^	21 (36) ^c^	175 (60)
Age ***						
Mean ± SD (range)	32.1 ± 17.4 (4–75) ^b^	42.5 ± 17.2 (5–80) ^a^	36.5 ± 16.5 (9–65) ^a,b^	38.6 ± 16.5 (15–65) ^a,b^	33.2 ± 33.2 (9–80) ^b^	36.6 ± 17.4 (4–80)
Occupation *n* (%)						
Farmer (including students)	88 (87)	68 (92)	31 (88)	23 (100)	54 (92)	264 (90)
Chapatti consumption (g/kg b.w./day)						
Mean ± SD (range) (g/ day) ***	324 ± 118 (0–510) ^b^	403 ± 120 (85–680) ^a^	369 ± 135 (0–595) ^a,b^	436 ± 82 (255–510) ^a^	341 ± 117 (170–510) ^b^	362 ± 123 (0–680)
Rice consumption (dry weight)						
Consumption rate (%)	50.5	43.2	31.4	13.0	46.8	42.5
Mean ± SD (range) (g/ day)	139 ± 69 (26–350)	146 ± 58 (24–234)	101 ± 28 (51–148)	174 ± 59 (106–212)	129 ± 54 (51–232)	136 ± 61 (24–350)

Statistical significance was determined based on *p* values: *p* < 0.05, *; *p* < 0.01, **; *p* < 0.001, ***; When there was an overall significant difference observed for the demographic factors between the villages, pair-wise comparison between the villages was made and villages were assigned to different groups a, b or c, following the order group mean a > b > c (a,b means not significantly different to either group a or b); the dry weight of rice consumption was derived from 29% of the rice consumption [24] and the rice consumption rate refers to the percentage of participants’ consumed rice on the previous day; the statistics for rice consumption were calculated only for those participants who consumed rice on the previous day.

**Table 4 toxins-14-00193-t004:** Mycotoxin concentrations for major mycotoxins found in rice and wheat samples (unit: µg/kg).

**Rice**						
Villages	Chak-46 (*n* = 8)	Chak-50 (*n* = 6)	BB (*n* = 17)	BP (*n* = 15)	KA (*n* = 16)	Total (*n* = 62)
AFB_1_ ** (*p* = 0.017)						
Positive *n*, %	3 (38)	0 (0)	13 (76)	9 (60)	16 (100)	41 (66)
Mean ± SD	15.01 ± 10.31	NA (i.e., <LOD)	13.29 ± 19.70	8.49 ± 14.74	1.09 ± 1.40	7.60 ± 14.06
Median (range)	17.89 (3.57–23.58)	nd	4.17 (0.18–71.56)	1.24 (0.31–36.88)	0.42 (0.17–5.26)	1.24 (0.17–71.56)
DON						
Positive *n*, %	0 (0)	0 (0)	0 (0)	0 (0)	0 (0)	0 (0)
FB_1_						
Positive *n*, %	0 (0)	2 (33)	0 (0)	0 (0)	1 (6)	3 (5)
Mean ± SD	NA	5.47 ± 2.74	NA	NA	NA	4.24 ± 2.87
Median (range)	nd	5.47 (3.53–7.41)	nd	nd	1.80	3.53 (1.80–7.41)
OTA						
Positive *n*, %	0 (0)	0 (0)	0 (0)	1 (7)	0 (0)	1 (2)
Mean ± SD	NA	NA	NA	0.40	NA	0.40
Median (range)	nd	nd	nd	0.40	nd	0.40
ZEN						
Positive *n*, %	8 (100)	6 (100)	17 (100)	15 (100)	16 (100)	62 (100)
Mean ± SD	6.91 ± 4.16	6.67 ± 3.79	4.77 ± 2.07	3.90 ± 2.88	5.43 ± 3.42	5.19 ± 3.19
Median (range)	5.75 (2.99–12.61)	6.51 (2.37–13.48)	4.22 (1.87–9.43)	2.65 (0.8–12.23)	4.45 (2.29–15.00)	4.13 (0.8–15.00)
**Wheat**						
Villages	Chak-46 (*n* = 40)	Chak-50 (*n* = 96)	BB (*n* = 13)	BP (*n* = 17)	KA (*n* = 29)	Total (*n* = 195)
AFB_1_						
Positive *n*, %	1 (3)	2 (2)	0 (0)	0 (0)	0 (0)	3 (2)
Mean ± SD	0.23	1.00 ± 0.83	NA	NA	NA	0.75 ± 0.74
Median (range)	0.23	1.00 (0.42–1.59)	nd	nd	nd	0.23 (0.23–1.59)
DON						
Positive *n*, %	0 (0)	0 (0)	0 (0)	0 (0)	0 (0)	0 (0)
Mean ± SD	NA	NA	NA	NA	NA	NA
Median (range)	Nd	nd	nd	nd	nd	nd
FB_1_						
Positive *n*, %	0 (0)	1 (1)	0 (0)	0 (0)	0 (0)	1 (0.5)
Mean ± SD	NA	2.72	NA	NA	NA	2.72
Median (range)	Nd	2.72	nd	nd	nd	2.72
OTA						
Positive *n*, %	2 (5)	6 (6)	1 (8)	0 (0)	1 (3)	10 (5)
Mean ± SD	0.65 ± 0.35	2.15 ± 1.96	0.40	NA	1.46	1.61 ± 1.65
Median (range)	0.65 (0.40–0.90)	1.52 (0.40–5.17)	0.40	nd	1.46	1.13 (0.40–5.17)
ZEN * (*p* = 0.047)						
Positive *n*, %	19 (48)	28 (29)	1 (8)	4 (24)	13 (45)	65 (33)
Mean ± SD	1.10 ± 0.47	1.57 ± 1.27	0.80	1.28 ± 0.56	1.10 ± 0.48	1.31 ± 0.92
Median (range)	0.80 (0.80–1.96)	1.22 (0.80–7.24)	0.80	1.25 (0.80–1.81)	0.80 (0.80–2.01)	0.80 (0.80–7.24)

nd: not detected, (i.e., <LOQ); NA: not available (i.e., no samples detected > LOQ); considering the low positive rate: mean, Standard deviation (SD), Median and range were calculated only for positive samples, excluding values <LOD; values between LOD and LOQ were replaced with ½ LOQ; Statistical significance was determined based on *p* values: *p* < 0.05, *; *p* < 0.01, **; *p* < 0.001, ***.

**Table 5 toxins-14-00193-t005:** Urinary mycotoxin biomarker levels in the studied population (Unit: pg/mL).

Villages	Chak-46 (*n* = 101)	Chak-50 (*n* = 74)	BB (*n* = 35)	BP (*n* = 23)	KA (*n* = 59)	Total (*n* = 292)
AFM_1_ ***						
Positive *n*, (%)	87 (86)	36 (49)	25 (72)	18 (78)	28 (48)	193 (66)
Mean ± SD	36.9 ± 58.1 ^a^	13.5 ± 30.1 ^b^	6.0 ± 6.9 ^b^	31.2 ± 34.6 ^a^	7.1 ± 15.5 ^b^	20.8 ± 41.3
Median (range)	17.4 (nd-420.8)	nd (nd-187.7)	2.7 (nd-27.9)	16.3 (nd-96.4)	nd (nd-101.6)	5.35 (nd-420.8)
DON ***						
Positive *n*, (%)	35 (35)	32 (43)	18 (51)	11 (48)	6 (10)	102 (35)
Mean ± SD	133.8 ± 241.3 ^b^	180.7 ± 417.4 ^a,b^	178.8 ± 184.5 ^a^	139.8 ± 113.1 ^a,b^	68.5 ± 66.8 ^c^	138.4 ± 266.5
Median (range)	nd (nd-2249.5)	nd (nd-3488.6)	102.6 (nd-755.0)	nd (nd-393.8)	nd (nd-416.6)	nd (nd-3490)
DOM-1 ***						
Positive *n*, (%)	12 (12)	28 (38)	6 (17)	3 (13)	13 (22)	61 (21)
Mean ± SD	49.0 ± 132.7 ^b^	64.4 ± 91.8 ^a^	35.1 ± 66.2 ^b^	46.2 ± 102.3 ^a,b^	53.1 ± 97.5 ^a,b^	51.8 ± 107.0
Median (range)	nd (nd-776.3)	nd (nd-380.7)	nd (nd-282.1)	nd (nd-387.7)	nd (nd-384.4)	nd (nd-776)
FB_1_ ***						
Positive *n*, (%)	89 (88)	56 (76)	33 (94)	20 (87)	26 (44)	225 (77)
Mean ± SD	15.6 ± 17.2 ^b^	18.6 ± 30.0 ^b^	30.9 ± 35.2 ^a^	27.9 ± 45.4 ^b^	6.60 ± 12.8 ^c^	17.4 ± 26.8
Median (range)	10.5 (nd-88.8)	10.9 (nd-170.6)	13.6 (nd-129.6)	16.8 (nd-225.2)	nd (nd-77.4)	9.3 (nd-225)
OTA ***						
Positive *n*, (%)	100 (99)	73 (99)	35 (100)	23 (100)	58 (98)	298 (99)
Mean ± SD	219.3 ± 505.4 ^a^	77.4 ± 84.8 ^b^	56.9 ± 54.4 ^b^	147.0 ± 162.7 ^a^	103.4 ± 76.6 ^a^	134.7 ± 312.0
Median (range)	106.1 (nd-4816.4)	60.6 (nd-611.5)	34.7 (1.4-210.6)	100 (13.6-704.3)	83.1 (nd-408.3)	74.5 (nd-4816.4)
ZEN ***						
Positive *n*, (%)	46 (46)	31 (42)	10 (29)	9 (39)	17 (29)	108 (37)
Mean ± SD	19.8 ± 29.9 ^a^	25.0 ± 79.1 ^a^	9.0 ± 7.5 ^a,b^	10.1 ± 7.4 ^a^	8.2 ± 9.0 ^b^	16.7 ± 44.1
Median (range)	nd (nd-174.5)	nd (nd-673.3)	nd (nd-32.0)	nd (nd-25.9)	nd (nd-55.4)	nd (nd-673.3)
α-ZEL **						
Positive *n*, (%)	35 (35)	21 (28)	5 (14)	7 (30)	11 (19)	79 (27)
Mean ± SD	11.5 ± 13.3 ^a^	13.2 ± 21.4 ^a^	6.6 ± 4.4 ^b^	9.6 ± 8.6 ^a^	8.2 ± 10.6 ^b^	10.5 ± 14.4
Median (range)	nd (nd-80.9)	nd (nd-160.8)	nd (nd-25.0)	nd (nd-33.7)	nd (nd-64.0)	nd (nd-160.8)
β-ZEL ***						
Positive *n*, (%)	43 (43)	34 (46)	15 (43)	12 (52)	8 (14)	111 (38)
Mean ± SD	25.5 ± 43.4 ^a^	18.2 ± 21.8 ^a^	24.9 ± 36.1 ^a^	17.0 ± 15.8 ^a^	11.0 ± 18.3 ^b^	20.0 ± 32.0
Median (range)	nd (nd-361.1)	nd (nd-111.2)	nd (nd-184.4)	10.0 (nd-62.1)	nd (nd-87.6)	nd (nd-361.1)

nd: not detectable (i.e., <LOD); a. The unit for all the listed values is pg/mL, ½ LOD were assigned to values below detection limit. Significant differences were observed for all the mycotoxin urinary biomarkers (*p* < 0.001), apart from α-ZEL (with a *p*-value of 0.002) which was observed for all the mycotoxin biomarkers across the five villages. Pair-wise comparisons between villages were made and villages were assigned to different groups a, b or c, following the order group a > b > c (a,b means not significantly different to either group a or b); Statistical significance was determined based on *p* values: *p* < 0.05, *; *p* < 0.01, **; *p* < 0.001, ***.

**Table 6 toxins-14-00193-t006:** Dietary exposure estimation with urinary biomarker approach.

Mycotoxin	Urinary Excretion Rate (%) ^b^	Mean Biomarker Conc (Max.) (µg/L)	PDI Mean (Max.) (µg/kg bw/Day)	Established TDI (µg/kg bw/Day) ^c^
AFB_1_	1.3	0.021 (0.421)	0.043 (0.852)	As low as possible
DON ^a^	70	0.140 (3.489)	0.006 (0.153)	1
FB_1_	0.3	0.017 (0.225)	0.159 (1.741)	2
OTA	2.5	0.135 (4.816)	0.145 (5.137)	0.017
ZEN ^a^	9.4	0.017 (0.673)	0.005 (0.214)	0.25

Daily urine volumes were assumed to be 36 and 18 mL/kg/day for children (4–10 years) and adolescents (10–19 years), and 2000 mL/day for adult males and 1.6 L/day for adult females, respectively, as recommended by EFSA FEEDAP Panel, (2012). ^a^. no data are available for the excretion rate of DOM-1, α-ZEL and β-ZEL, therefore the dietary exposure of DON and ZEN were estimated by urinary DON and ZEN, respectively; ^b^. the references for the urinary excretion rate of the mycotoxin urinary biomarkers are as follows: AFM1, 1.3% [29]; DON, 70% [30]; FB1, 0.3% [31]; OTA, 2.5% [32]; ZEN, 9.4% [33]; ^c^. the TDI defined by the Joint FAO/WHO Expert Committee on Food Additives (JECFA) for DON, OTA and FB1 are used [25,26,27]; the TDI defined by the European Food Safety Authority (EFSA) for ZEN is used as reference [28].

## Data Availability

Not applicable.

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
