# Peer review of "Super-Sensitive LC-MS Analyses of Exposure Biomarkers for Multiple Mycotoxins in a Rural Pakistan Population"

_toxins, 2022, doi:10.3390/toxins14030193_

Round 1

Reviewer 1 Report

The present study describes the multi-mycotoxin methods for the analysis of food and urine samples, and provides the risk assessment of dietary mycotoxin exposure of rural Pakistan population. Although the research study might be promising, the scientific contribution in its present state is weak and some major issues need to be addressed. Methods should be more clearly described, the method validation clarified, and numerous linguistic and typographical mistakes corrected. The manuscript has some merits in the aspects of the risk assessment but the overall impact of the study is limited. 

Some specific comments are given below:

Line 83. Please clarify why are the urine QC samples included in table 1.

Line 87-89. Please clarify the obtained results for the repeatability, were both extraction and analysis for the 26 samples performed twice or it was just a reinjection of the same extract?

Line 90. Table 1. requires some clarifications. Namely, according to the title it relates to the food analysis, but the authors state the use of urine QC samples to check the recovery. Could the authors please clarify that? Moreover, to what matrix do the matrix LOD and matrix LOQ refer to, wheat or rice? Why linearity, intra-day and inter-day variations are not included in Table 1?

Line 107. Table 2. Why is the lowest calibration standard below LOQ? Why is the LOD included in the calibration range?

Line 139 – 140. Please specify measured concentrations for the FB1 and OTA in rice samples.

Line 145 – 147. Please specify measured concentrations for the FB1 and OTA in wheat samples.

Line 101. Repeated samples showed significant differences for the urinary DON level. Did the analyst consider the measurement of uncertainty value? Please indicate the mu value for each target analyte for both methods.

Line 102 – 106. Do the samples compared belong to the same cohort?

Line 368 – 375. Under the section reagents and chemicals, the authors list only standards and cartridges. Please indicate the purity of standards used. Moreover, the specifications of other materials used should be mentioned here as well, such are solvents (including the grade of solvents), β-glucuronidase, etc. In addition, preparation of reference material, calibration standards, ISTD solution needs to be included.

Line 385 – 387. The collection of food samples is poorly described. The amount of samples collected per household / local shop should be indicated, as well as the type of wheat collected, household storage conditions, for rice whether it was bulk or packed rice, etc.? The total number of samples collected for each sample category, as well as the period (months and year(s)) when the samples were collected, should be mentioned. Moreover, once samples were collected, how were they stored until analysis?

Line 387 – 388. The authors indicate that this is an extension of the earlier study. If the same samples from the earlier study (n=395) were used in this one (n=292), the selection criteria should be included. Also, the collection period, including months and year(s) should be included.

Line 396 – 408. Sample preparation for food mycotoxin analysis should be described in more detail, e.g. temperature at which extraction was performed, centrifuge speed and temperature, were the food samples also spiked with ISTDs, etc. Furthermore, although it is mentioned that the sample was diluted, the dilution factor is missing.

Line 411. Please specify the speed and the temperature of the centrifuge when used for urine sample preparation.

Line 426. I have some serious concerns regarding the method validation for the analysis of food samples. Do the QC levels indicated relate to the method for the analysis of food samples or the method for urine sample analysis? How were the QC samples prepared, and why are they expressed as ng/mL while the calibration standards are expressed as μg/kg? What is the matrix effect? Did the analyst use matrix-matched calibration standards? How were recovery and repeatability calculated, what was the intra-day and inter-day variation for the method used for food mycotoxin analysis? Finally, the concentration of each calibration standard should be indicated as the calibration range mentioned in table 1. doesn’t really seem fit for purpose, i.e. how many calibration points were included between LOQ and 20 μg/kg where most of the results fit in? What is the linearity in that range?

Author Response

Dear reviewer,

Thank you for your comment and please see the attached file for our responses to your comment.

Best regards,

Reviewer 2 Report

The manuscript measures multiple mycotoxin biomarkers of Aflatoxin B1 (AFB1), deoxynivalenol (DON), fumonisin B1 (FB1), ochratoxin A (OTA), and zearalenone (ZEN) in urine samples of the Pakistan population from rural areas using a sensitive LC-MS/MS method. The approach used in this study was effective in conducting exposure assessment and risk management in the studied regions. The analysis of multiple mycotoxins showed that populations from the Punjab region of Pakistan had a high prevalence of AFB1 and OTA exposure. The estimated AFB1 and OTA exposure levels were comparable to those countries, but the pharmacokinetics for OTA and ZEN is not well characterized. The obtained results about foods and population contamination by mycotoxin are relevant to Pakistan, but the analytical methodology is known and some correlations among biomarkers and Aflatoxins need more research to be clarified. The biomarkers could be identified in the abstract section.

Author Response

(The authors gave the same response as above.)

Reviewer 3 Report

High levels of mycotoxin contamination have been reported in various food commodities in Pakistan. However, there lack of the method of exposure assessment study using multiple mycotoxins biomarker in humans. This study has simultaneously assessed the exposure of the five major mycotoxins: aflatoxin B1, deoxynivalenol, fumonisin B1, ochratoxin A and zearalenone exposure in a Pakistani population using an integrated approach of human biomonitoring. This is a interesting topic. The experiment was well-designed and the data was well presented.

  1. Line 11, please correct “(N=292)”to “(N = 292)”. Please check the similar issues throughout the manuscript.
  2. Lines 11-12, please add the full name for “LC12 MS/MS”when it was firstly appeared in the manuscript. Please check the similar issues throughout the manuscript.
  3. Line 18, please correct “TDI=17”to “TDI = 17”.
  4. Line 36, please correct “animals .”to “”. 
  5. Lines 41-45, Please add some new reference about the occurrence of mycotoxins in cereals. Such as 1)  Individual and Combined Occurrence of Mycotoxins in Feed Ingredients and Complete Feeds in China. Toxins, 2018; 2) Occurrence of Aflatoxin B1, deoxynivalenol and zearalenone in feeds in China during 2018–2020; Journal of Animal Science and Biotechnology 2021; 3) Invited review: Remediation strategies for mycotoxin control in feed.  Journal of Animal Science and Biotechnology 202.
  6. Table 1, what is the RAmeans? Similar to QC, CV, AFB1, etc. Please add the full name for all the abbreviation have been used in the table. Please check the similar issues throughout the manuscript.
  7. Line 99, please correct “(n=31)”to “(n = 31)”. There are many similar issues. Please check the similar issues throughout the manuscript.
  8. Line 105 please correct “(p>0.05)”to “(p > 05)”. There are many similar issues. Please check the similar issues throughout the manuscript.
  9. Line 124, Table 1? Please use the superscript for the letters of “a”, “b”, etc. to indicate the difference. Please check the similar issues throughout the manuscript.
  10. Line 148, Table 2?
  11. Line 220, please correct “a)”to “(a)”.  

Author Response

(The authors gave the same response as above.)

Reviewer 4 Report

It is a really interesting research, probably for further research should be included other food products, other cereals, pseudocereals, and milk. 

Line 36,67, 249, 437: extra space

Line 73-75: For each food product should add the specific reference.

Line 87, 473: p-value should be in italic

Line 151: mistake super-sensitive

Line 385: 24h, no extend format

Line 408: mistake software

Figure 1a – the legend of different colours should be included

Line 229: missing a point in final of sentence

Author Response

(The authors gave the same response as above.)

Reviewer 5 Report

In this paper, the author aimed to simultaneously assess the exposure of the five major mycotoxins of AFB1, DON, FB1, OTA and ZEN in the Pakistani population using an integrated approach of human biomonitoring, by collecting and analyzing the human urine samples (N=292) and food samples (Rice and wheat). The manuscript is well written but the title of the article does not match the contents. I think there are hundreds of similar papers in different journals. In my opinion, the manuscript does not give any novel to the knowledge on this topic. I'm sorry.

Furthermore, I have some comments regarding the methodology, that is the sample (urine and food samples) collection method, and relationship between the collected human urine samples and the dietary intakes of individuals (food samples).

Author Response

(The authors gave the same response as above.)

Round 2

Reviewer 1 Report

In the revised version of the manuscript, the authors have addressed most of my concerns. However, some revision is still needed:

Specific comments are given below:

  • Line 88-90 Inter-day variation should refer to different days while intra-day variation is variation within one day. Also, please specify which standard concentration was used to assess both intra- and inter-day variations?
  • Table 1 Are the intra- and inter-day variations for rice and wheat samples the same? If not, please specify for each as well as the mycotoxin concentration used.
  • Table 1. The clarification to what matrix do the matrix LOD and matrix LOQ refer to, wheat or rice is still not provided?
  • Table 4. does not seem to be correct. For example, rice FB1 concentration in Chak-50 ranges from nd to 7.41 with 2 positive samples. To achieve the mean value of 2.04 the value for the second positive sample should be negative. Also, the total mean for the FB1 in the rice sample is 0.510 which is way below LOQ. In addition, nd is not correctly used throughout the table, i.e. if nd is defined as not detected (<LOQ) than the same abbreviation cannot be used for mean. Perhaps for the mean, NA (not applicable) should be used instead. Moreover, the authors calculate the mean for 2 positive samples but not for 3 or 10 positive samples. Why is that? Finally, please specify what do the letters a and b after mean values for ZEN in wheat refer to.
  • It is recommended to standardize the result presentation in Tables 4 and 5. Namely in Table 4. The % of positive samples is in the brackets and the number of positive samples in front, while it is the opposite in Table 5.
  • While the information on the PDI obtained from the urinary biomarkers is extremely valuable, it would be interesting to compare this with the risk assessment based on the presence of mycotoxins in food. Namely, based on the publicly available information on the average consumption of rice/wheat and an average person's body weight in Pakistan, the EDI for each mycotoxin can be calculated as well as MoE at BMDL01, BMDL05 and BMDL10 for carcinogenic compounds.
  • Line 496 to what mycotoxin the concentration 9.6 μg/kg refers to? OTA?

Author Response

Dear reviewer,

Thank you for your comments, please see the attachment for our responses to your comments.

Reviewer 5 Report

The paper has been greatly improved, and can be accepted in present form.

Author Response

Dear reviewer,

Thank you for the positive comments on our revised manuscript.

Best regards,

Lei Xia

Round 3

Reviewer 1 Report

The authors have addressed most of my comments. Still there are two minor comments regarding table 4:

  • Although, the approach for assigning half the LOD value for the negative samples is extremely valuable for the risk assessment studies, here using that approach to calculate median and mean might be very confusing for the reader. Thus, if authors prefer to use all positive samples for the calculation of mean and median, I would suggest that for all positive samples with the mycotoxin concentration >LOD but <LOQ to assign the value of concentration half the LOQ, while to omit those with values <LOD in the calculations of median and mean. Also if using only positive samples, below the table should be indicated: Values between LOD and LOQ were replaced with ½ LOQ; Mean is calculated in positive samples, excluding values < LOD
  • If there are 6 and 10 positive wheat samples for the OTA, using the suggestion above you should be able to provide the information for the mean and median for positive samples.

Author Response

Dear reviewer,

Thank you for your suggestion, we have updated the table accordingly and please see the attached file for detail.

Best regards,

Lei Xia
